# Heat Generation at the Implant–Bone Interface by Insertion of Ceramic and Titanium Implants

**DOI:** 10.3390/jcm8101541

**Published:** 2019-09-25

**Authors:** Holger Zipprich, Paul Weigl, Eugenie König, Alexandra Toderas, Ümniye Balaban, Christoph Ratka

**Affiliations:** 1Department of Prosthodontics, Faculty of Oral and Dental Medicine at Goethe University, 60590 Frankfurt am Main, Germany; weigl@em.uni-frankfurt.de (P.W.); c.ratka@gmx.de (C.R.); 2Private Practice, 60385 Frankfurt am Main, Germany; koenig-@hotmail.de; 3Private Practice, 60313 Frankfurt am Main, Germany; toderasalexandra@gmail.com; 4Institute of Biostatistics and Mathematical Modelling at Goethe University, 60590 Frankfurt am Main, Germany; balaban@med.uni-frankfurt.de

**Keywords:** zirconia, dental implant, insertion, bone–implant interface, heat, bone damage, early loss

## Abstract

Purpose: The aim of this study is to record material- and surface-dependent heat dissipation during the process of inserting implants into native animal bone. Materials and Methods: Implants made of titanium and zirconium that were identical in macrodesign were inserted under controlled conditions into a bovine rib tempered to 37 °C. The resulting surface temperature was measured on two bone windows by an infrared camera. The results of the six experimental groups, ceramic machined (1), sandblasted (2), and sandblasted and acid-etched surfaces (3) versus titanium implants with the corresponding surfaces (4, 5, and 6) were statistically tested. Results: The average temperature increase, 3 mm subcrestally at ceramic implants, differed with high statistical significance (*p* = 7.163 × 10^−9^, resulting from group-adjusted linear mixed-effects model) from titanium. The surface texture of ceramic implants shows a statistical difference between group 3 (15.44 ± 3.63 °C) and group 1 (19.94 ± 3.28 °C) or group 2 (19.39 ± 5.73 °C) surfaces. Within the titanium implants, the temperature changes were similar for all surfaces. Conclusion: Within the limits of an in vitro study, the high temperature rises at ceramic versus titanium implants should be limited by a very slow insertion velocity.

## 1. Introduction

Although titanium and its alloys have been used successfully in dental implantology for more than five decades, there is increasing demand for a nonmetallic alternative.

The alternative material should, of course, retain the good properties of titanium and, if possible, even improve it and eliminate the disadvantages. These primarily include the color, which can cause a greyish discoloration of the peri-implant soft tissue in the esthetic area, leading to the patient’s perception that the replaced tooth root is made of metal. Patients particularly prioritize high simulation quality for materials used to replace lost tissue. This has led to a similar development in restorative dentistry, although inlays and onlays made of gold alloy or crowns and bridges with a metal framework have had excellent long-term clinical results. Nevertheless, their importance is increasingly diminishing because they are being replaced by all-ceramic restorations, as patients perceive ceramic to simulate tooth hardness much better than metal. Even metal crowns and bridge frameworks that are nonvisible, because they are fully veneered, have increasingly been replaced by high-strength ceramic versions. This trend, which exists in restorative dentistry, now seems to be repeated in oral implantology: an artificial tooth root made of ceramic simulates the replaced tissue better than titanium. In other words, in terms of the choice of materials, patients always opt for, or clinicians always tend to use, ceramics for replacing tooth tissue.

A major disadvantage of ceramic implants is their early loss that occurs in clinical trials and is observable in daily practice, which is usually associated with no clinically apparent inflammation. This phenomenon, described as aseptic loosening, is caused by a lack of osseointegration [1] or linked to reactivation of the inflammatory immune system [2]. Different forms of bone injury caused by osteotomy can be eliminated because identical osteotomy protocols are used clinically to create the bone cavity for titanium and ceramic implants. Only during implant insertion can it be assumed that there is a higher risk of overheating with ceramics than with titanium because of their different thermal conductivities. Overheating leads to increased damage or destruction of the bone directly on the implant surface.

The aim of this study is to record the material-dependent heat dissipation during the process of inserting implants into native animal bone in an in vitro setup. In addition, the influence of different implant surface treatments is evaluated. Implants made of titanium and zirconium that were identical in macrodesign were used for this purpose.

## 2. Materials and Methods

### 2.1. Implant Specimen and Surface Treatments

Y-TZP (Yttria–Tetragonal Zirconia Polycrystal) two-piece zirconia implants (5 × 14 mm) from the BPI system (BPI Biological and Physical GmbH & Co. KG, Sindelfingen, Germany) served in their packed original state as the specimen for machined ceramic implants. These cylindrical screw implants were designed with four apical cutting grooves and had a self-cutting property (Figure 1).

However, the implants made of titanium from the same manufacturer differed in macrodesign from the above-mentioned ceramic implants. Therefore, the endosteal parts of the BPI ceramic implants were re-engineered to match endosteal titanium grade 4 machined parts. To simplify the re-engineering, the titanium implants were not manufactured as a two-piece (implant and abutment separable) but as a one-piece implant system (Figure 1).

In addition to the influence of the implant materials on heat generation at the implant–bone interface during the insertion process, the effect of an implant surface modification was also investigated. The topography of each surface type was scanned under a scanning electron microscope (SEM S-4500, Hitachi, Tokyo, Japan) (Figure 2). The roughness (Sa) of the implant surface in each group was measured according to ISO-25178 by a laser-scanning microscope (KEYENCE, VK-X100, Osaka, Japan).

Based on two different methods of surface treatment, 6 groups of implant specimens were produced:

Groups 1 and 4: nontreated, machined, Y-TZP, two-piece zirconia (group 1) and one-piece titanium implants (group 4). The machined surface is shown in Figure 2 (Groups 1 and 4). The diameter of both implants was 5.00 mm.

Groups 2 and 5: once testing with group 1 and 4 implant specimens was finished, each implant was cleaned by water steam jet and 70% ethanol. The implant was additionally given a microstructure by sandblasting with corundum (mesh size = 110 µm, working pressure = 6 bar) at an automatic turning speed of 100 rpm until the diameter of the implants was reduced to 4.95 mm. The SEM images in Figure 2 (Groups 2 and 5) show the result of the surface treatment.

Groups 3 and 6: Once testing with group 2 and 5 implant specimens was finished, each implant was cleaned again by water steam jet and ethanol (70%). The sandblasted surface was then acid-etched to produce a nanostructure showing hydrophilic properties. The result of the acid-etching process is shown in Figure 2 (Groups 3 and 6). The diameter of the implants was further reduced to 4.90 mm.

### 2.2. Measurement of Heat Generation and Dissipation

The temperature at the bone–implant interface during insertion was dynamically measured by an infrared camera PI160 (PI16048T900, Optris, Berlin, Germany). The infrared camera took real-time images at a frequency of 120 Hz with an uncooled microbolometer focal plane array (FPA) detector with 160 × 120 pixels. The sensor had a spectral range from 7.5 to 13 μm and was equipped with a wide-angle lens with a field of view (FOV) of 48° × 37°. The thermal sensitivity with respect to the relative accuracy of the camera was 0.1 °C, and the minimum measuring distance was 20 mm. The absolute accuracy of the measurement was 2 °C. The temperature data from the camera were transmitted to the computer as thermal images, evaluated using Optris PI Connect software, and transferred to a spreadsheet for statistical analysis.

### 2.3. Bone Segments

It has been reported that the macroscopic structure of cortex and cancellous bone in the bovine rib is similar to that in the human jaw, and the bone density dependent on its position could be comparable to classes I–IV according to Lekholm & Zarb or D2–D4 according to the Mish classification in the human jaw [3,4,5,6]. The distal part of the bovine rib was used and classified as D2/D3 bone. Therefore, 14 ribs were used in this study to simulate the condition in the human jaw. Bone density measurements have not been made because the friction when inserting cylindrical implants arises mainly on the hard cortex bone [7,8] if a sufficiently high torque has been generated there (40 to 60 Ncm in this study). Therefore, when selecting the bone segments, particular care was taken to ensure that the cortical layer thickness was somewhat the same for all bovine bony ribs [7].

### 2.4. Experimental Settings

In each experiment, the implant was connected to the screwing unit, ensuring a stable rotational speed during insertion of the implant. In the screwing unit, the driving force was produced by a pneumatic cylinder (C85N10-75, ISO air cylinder series C85, SMC Pneumatik GmbH, Egelsbach, Germany). Then, the motor (DC-servo drive series 3564K024B CS, Faulhaber GmbH & Co KG, Schöneich, Germany) connected to a gearbox (series 30/1, 134:1, Faulhaber GmbH & Co KG, Schöneich, Germany) transmitted the force to the torque sensor (DR-20-2Nm; Lorenz Messtechnik GmbH, Alfdorf, Germany) and the collet holder. The implant was connected to the collet holder by a round-headed Allen key (3 mm) to allow rotation. The Allen key was loosely fitted to the mounting post of the implant so that insertion of the implant could remain centered in the prepared bone cavity guided by implant threads and a cone-shaped apex.

The screwing unit could be driven forward or backward axially, and its movement was recorded by a linear inductive distance sensor (LVP 100, Micro-Epsilon Messtechnik GmbH & Co. KG, Ortenburg, Germany). To simulate the environment of surrounding tissues, the bone segment and the implant were covered in a preheated polycarbonate thermo box (37 °C) to prevent heat dissipation. The infrared camera was fixed to the cover of the thermo box. The complete experimental settings are shown in Figure 3.

### 2.5. Experimental Protocol

Fourteen of the zirconia implants (groups 1–3) were included in the whole experiment, and each implant underwent 3 loops as a result of the surface treatments applied. After each implant had been assigned to one bovine rib, each bovine rib (BR_(1)_–BR_(14)_) was cut into 3 sequential rib segments (RS1_BR(1)_–RS3_BR(1)_ to RS1_BR(14)_–RS3_BR(14)_). Each implant (I1–I14) would undergo one type of surface modification (machined (m), sandblasted (sb), sandblasted and acid-etched (ae)) before each test program (TP1–TP3). The test sequence of the samples and the corresponding rib segments were organized as follows:

Each implant was inserted into one new rib segment after each surface modification. First, all machined implants (I1_(m)_–I14_(m)_) were inserted into 14 rib segments (RS1_BR(1)–BR(14)_). Then, after sandblasting, implants (I1_(sb)_–I14_(sb)_) were inserted into rib segments (RS2_BR(1)–BR(14)_). Finally, after acid-etched treatment, implants (I1_(ae)_–I14_(ae)_) were inserted into rib segments (RS3_BR(1)–BR(14)_). Surface modification usually took several days. In order to maintain the bone in the same fresh condition in all the tests, bone segments were frozen beforehand in Ringer’s solution at −10 °C and were defrosted immediately prior to the experiment.

The titanium implants underwent identical protocols. Since they were made especially for this study, a higher number of pieces was available at the beginning of the trial. Therefore, with torques outside the interval of 40 to 60 Ncm, new Ti implants were used to repeat the insertion test so that 15 measurements per group could be realized (Table 1). The selected torque window was clinically sufficient for immediate restoration of ceramic implants (>35 Ncm), which are often still used as one-piece implants. Furthermore, smaller torques do not produce sufficiently high temperatures by friction between the implant surface and the hard cortical bone [8] to be able to detect the effect of different thermal conductivities between titanium and ceramic at the bone–implant interface.

Contrary to Ti implants, for this study only 14 ceramic implants were available from the manufacturer. Therefore, when the torque was too low or too high, bone penetration tests with the respective surface structure were no longer repeated in order to mechanically protect the connection of the implant driver with the ceramic implant. In addition, cleaning the surfaces of bone remnants would have been a process in which, above all, the roughened ceramic surfaces could have been structurally changed; thus, the equality of specimens could no longer be ensured. Because of this limitation of available ceramic implants, only 8 measurements in group 1, 11 measurements in group 2, and 11 measurements in group 3 could be realized (Table 1).

When one test started, the selected bone segment would be defrosted in Ringer’s solution at 37 °C and stored at the same temperature in the oven to simulate the human condition.

The implant bed was prepared using a twist drill (Type N, 118° DIN 338 R-N, Gühring KG, Albstadt, Germany) at the speed of 800 rpm. Since the bone was already wetted, additional water cooling was not necessary. To ensure a comparable insertion resistance in the three loops, the diameter of each implant site matched the implant diameter, that is, twist drills with diameters of 4.3, 4.25, and 4.20 mm were used in loops 1, 2, and 3, respectively.

To dynamically record temperature directly at the bone–implant interface, two bone windows perpendicular to the insertion path were made at 3 and 9 mm subcrestally by a twist drill (Figure 4).

Before each test, the bone was brought to a temperature of 37 °C inside an oven with saturated humidity. After the bone segment was immobilized, the implant was prewetted in Ringer’s solution used to simulate human blood, and then it was driven with a starting pressure of 5 N and a constant rotational speed of 25 rpm to a depth of 12 mm inside the bone. Meanwhile, the infrared camera continuously recorded the temperature through each bone window (2b & T7 in Figure 5), and the difference between the maximum and the starting temperature (37 °C) during insertion would be calculated (∆T). The change of torque was recorded by torque sensors (DR-20, Lorenz Messtechnik GmbH, Alfdorf, Germany) with the frequency of 100 Hz. Data from groups in which the insertion torque was above 60 Ncm or below 40 Ncm were excluded from the analysis.

After each experiment, the thickness of cortical bone at the insertion point was measured with a caliper gauge. When the temperature, measured at the position 3 mm subcrestally, exceeded 47 °C (∆T >10 °C) [9], 50 °C (>13 °C) [10], or 55 °C (>18 °C) [11], the corresponding exposure time was calculated.

### 2.6. Statistical Analysis

Continuous variables are represented as mean ± standard deviation for each group. Comparisons were performed using the Mann–Whitney *U* test, *t*-test, or paired *t*-test. Comparisons with more than 2 groups were analyzed by ANOVA or the Kruskal–Wallis test, depending on Gaussian distribution. Additionally, comparisons using adjusted linear mixed-effects models (LMEs) were performed. Gaussian distributions of data were assessed with the Shapiro–Wilk test. The level of significance in post hoc tests was corrected for multiple testing. The level of significance was set at α = 0.05, and all tests were two-sided. Statistical analysis was performed using R 3.6.1 with the packages multicomp 1.4-10, nlme 3.1-140, and plotrix 3.6-3. R, and the packages used are available from CRAN at http://CRAN.R-project.org/.

## 3. Results

### 3.1. Group-Related Parameters of Specimens and in Vitro Setup

To ensure that temperature change assessments were comparable with each other, only the implants with insertion torque between 40 and 60 Ncm were included in the analysis. Six machined and three sandblasted implants (groups 1 and 2, respectively) that exceeded an insertion torque of 60 Ncm, and three sandblasted etched implants (group 3) that had an insertion torque under 40 Ncm, were excluded from the analysis. From groups 4 to 6, all 15 specimens were analyzed. The resulting insertion torque was comparable for all 6 groups (group 1: 52.61 ± 5.10 Ncm; group 2: 48.97 ± 6.43 Ncm; group 3: 48.79 ± 5.19 Ncm; group 4: 50.10 ± 8.69 Ncm; group 5: 50.99 ± 7.97 Ncm; group 6: 50.72 ± 7.22 Ncm; *p* > 0.05; Figure 5a.).

The thickness of the cortical plate of the bovine bone rib was similar between the 6 groups (group 1: 2.51 ± 0.25 mm; group 2: 2.75 ± 0.47 mm; group 3: 2.61 ± 0.39 mm; group 4: 2.46 ± 0.12 mm; group 5: 2.46 ± 0.12 mm; group 6: 2.42 ± 0.09 mm; *p* > 0.05; Figure 5b).

The average roughness (Sa) of the machined implants (group 1: 0.93 ± 0.16 μm; group 4: 0.37 ± 0.04) was, as expected, significantly lower than that of sandblasted implants (group 2: 1.86 ± 0.38 μm; group 5: 2.15 ± 0.24 μm; all comparisons have *p*-values < 0.001) and sandblasted and acid-etched implants (group 3: 2.14 ± 0.62 μm; group 6: 1.89 ± 0.18 μm; all comparisons have *p*-values < 0.001; Figure 5c). The difference in surface roughness between ceramic and titanium implants after sandblasting (group 2 vs. group 5) and after sandblasting and acid-etching (groups 3 vs. 6) was not statistically significant (*p* > 0.05). The roughness between the machined surfaces (groups 1 and 4) was not statistically significant (*p* > 0.05). In paired comparisons of the surface treatments, the machined and the sandblasted groups showed significant differences (groups 1 and 4 *p* = 1.213 × 10^−4^; groups 2 and 5 *p* = 0.025756). The sandblasted and acid-etched groups (groups 3 and 6) showed no significant differences (*p* > 0.05).

### 3.2. Maximum Temperature Increase

#### 3.2.1. Bone Window 1

In bone window 1 (located 3 mm subcrestally), the average temperature increase at the zirconia implants had a high statistically significant difference to that of titanium implants (*p* = 7.163 × 10^–9^, resulting from group-adjusted linear mixed-effects models).

Within the zirconia implants (groups 1: 19.94 °C ± 3.28 °C; 2: 19.39 °C ± 5.73 °C; 3: 15.44 °C ± 3.63 °C) there were significant differences between groups 2 and 3 (*p* = 0.04426) and groups 1 and 3 (*p* = 0.007525; Figure 6). By contrast, there was no significant difference between the comparison of groups 1 and 2 (*p* > 0.05).

Within the titanium implants (group 4: 9.33 °C ± 4.18 °C; group 5: 9.20 °C ± 3.31 °C; group 6: 7.68 °C ± 2.68 °C), the temperature changes were similar for all groups (*p* > 0.05).

In paired comparisons of the different materials with the identical surface treatments, all groups showed highly significant differences (groups 1 and 4 *p* = 3.728 × 10^–11^; groups 2 and 5 *p* = 7.721 × 10^–6^; groups 3 and 6 *p* = 1.725 × 10^–6^
Figure 6).

#### 3.2.2. Bone Window 2

In bone window 2 (located 9 mm subcrestally), the average temperature increase at the zirconia implants had a high statistically significant difference than that of titanium implants (*p* = 7.182 × 10^–12^, resulting from group-adjusted linear mixed-effects models).

Within the zirconia implants (group 1: 7.50 °C ± 2.64 °C; group 2: 6.14 °C ± 2.40 °C; group 3: 6.07 °C ± 3.11 °C), the temperature changes were similar for all groups (*p* > 0.05).

Within the titanium implants (group 4: 6.03 °C ± 3.69 °C; group 5: 4.33 °C ± 3.26 °C; group 6: 4.41 °C ± 2.47 °C), the temperature changes were similar for all groups (*p* > 0.05).

### 3.3. Overheating Exposure Time

#### 3.3.1. Bone Window 1

ΔT > 10 °C

In bone window 1, the average exposure times under ∆T > 10 °C at the zirconia implants were 39.06 s for machined (all 8 of 8 in group 1), 28.10 s for sandblasted (11 of 11 in group 2), and then decreased to 20.37 s for sandblasted etched implants (10 of 11 in group 3). The average exposure times under ∆T >10 °C at the titanium implants were 22.49 s for machined (7 of 15 in group 4), 23.05 s for sandblasted (6 of 15 in group 5), and then decreased to 17.45 s for sandblasted etched implants (2 of 15 in group 6). None of the tested implants experienced a temperature increase of more than 10 °C over a period of more than 60 s.

ΔT > 13 °C

The average exposure time under ∆T >13 °C for ceramic implants was 25.10 s in group 1, 16.93 s in group 2, and 10.19 s in group 3. Three machined implants (group 1: 42.2, 36.1, and 32.1 s) and two sandblasted implants (group 2: 34.6 and 33.2 s) experienced more than 30 s temperature increase of over 13 °C. No sandblasted and acid-etched implants (group 3) experienced such a temperature change for longer than 30 s.

The average exposure time under ∆T >13 °C for titanium implants was 19.03 s at 3 of 15 implants in group 4 (group 4: 19.8, 21.8, and 15.6 s). Two machined implants experienced more than 13 °C temperature increase (group 5: 13.9 and 17.4 s) No sandblasted and acid-etched implants (group 6) experienced a temperature increase of 13 °C or more.

ΔT > 18 °C

The average exposure time under ∆T > 18 °C for ceramic implants was 11.36 s at 5 of 8 implants in group 1, 7.98 s at 6 of 11 implants in group 2, and 0.43 s at 3 of 11 implants in group 3. Two machined ceramic implants (group 1: 25.4 and 14.2 s) and two sandblasted ceramic implants (group 2: 19.0 and 16.7 s) experienced a temperature increase of more than 18 °C for more than 14 s. No sandblasted and acid-etched implants (group 3) experienced a temperature change greater than 18 °C for more than 14 s.

Only one titanium implant (group 4) exceeded 18.70 °C and had an exposure time of 20.40 s for ∆T > 18 °C.

#### 3.3.2. Bone Window 2

In bone window 2, the temperature increase never exceeded values > 10 °C.

### 3.4. Material-Dependent Thermal Energy Propagation

Figure 7 shows the image captured by the infrared camera during the process of inserting titanium and ceramic implants, both with sandblasted and acid-etched surfaces. Prior to the insertion process—meaning the implants were completely outside of the bone cavity—there was very little temperature difference between the titanium implant (33.2 °C) and the ceramic implant (31.2 °C). Bone window 1 shows the nearly regulated temperature of the thermo box (Figure 3) encapsulating the in vitro setup (37.5 °C; 37.2 °C).

At the moment of maximum temperature development during implant insertion, the ceramic implant was heated to 48.7 °C below the cortical bone (bone window 1) and to 33.6 °C outside the bone. In contrast, the titanium implant showed 42.9 °C within bone window 1 and 38.0 °C outside the bone. Thus, at the same distance, the titanium implant had a 3 times lower temperature gradient than the ceramic implant during the insertion process (ΔT_ceramic_ = 15.1 °C vs. ΔT_titanium_ = 4.9 °C).

The summary of all results is shown in Table 1.

## 4. Discussion

The aim of the in vitro study was to dynamically detect heat development at the bone–implant interface during the entire process of inserting a cylindrical screw implant into bone. For this purpose, an in vitro setup was developed, which achieves the highest possible quality of simulation of the clinical conditions and uses of test specimens, which differ in only one parameter, as far as possible.

### 4.1. Identical Specimen

It was important for good reproducibility of the temperature values at the implant surface that the implant specimens had an identical endosteal macrodesign and were identically manufactured (Figure 1 and Figure 2). The measured surface roughness (Figure 5b) did not differ significantly between the ceramic (ZrO_2_) and titanium implant materials in the sandblasted and sandblasted-etched surfaces. Only on the machined implants was the titanium surface statistically significantly smoother.

Repetition of the test series with the same implants after each additional surface treatment allowed for a nearly identical macrodesign with a different microdesign. Only removal of material by subtractive surface treatment caused a reduction in implant diameter. However, this was recorded by measurement and taken into account in the osteotomy protocol. In other words, the last cutter for bone preparation was also correspondingly smaller in diameter.

### 4.2. In Vitro Setup

Several published articles reporting on heat development when screwing an implant into the bone cavity have used bovine bone [12], in particular the bovine rib [9,10,13,14,15,16]. Bone segments with a cortical thickness of around 2.5 mm were chosen in order to mimic the known thickness of the human mandible (1.0–2.5 mm) [17,18,19] and to ensure the almost same amount of friction between bone and implant surfaces [7,8]. The temperature rise was generated in the ceramic implant mainly in the cortex bone (see bone window No. 1). In cancellous bone, however, there were no statistically significant differences in the temperature increase. This reinforces the assumed mechanism of temperature generation: in the case of a sufficiently high torque (40–60 Nm) and a conventional screw thread, most of the friction is generated between hard bone and the implant surface. Only special aggressive threads—for example, in Nobel Active in spongy bone, a mechanism other than friction—can generate high torque or primary stability—compression of trabeculae. Therefore, in the selection of the bone ribs, particular attention was paid to a comparable layer thickness of the cortex [7]. Since the spongy part of the rib in the present macrodesign of the examined implants (cylindrical, conventional screw thread) hardly generates any friction or heating, determination of the bone mineral density was omitted. Therefore, a high variation in the bone density in the spongy region of the rib probably has no influence on the measurement results in bone window 2.

Although blood circulation was absent in this case, the remaining conditions such as the chemical composition, density, humidity, structure, as well as mechanical and thermal properties were similar to those of the human bone. By contrast, artificial bone models cannot properly simulate the human bone because there are huge differences in thermal conductivity and heat capacity [20].

The anatomical structure—cortical bone and the underlying cancellous bone—are suitable for simulation of a toothless, healed alveolar bone. In order to keep the high anatomical variance of bovine ribs reasonably small in this investigation, the ribs were frozen after a test run so that measurements could at least be done on the identical bovine rib after the respective surface treatment of the implants for the second and third test runs.

An infrared camera (Figure 3) measured the temperature during the insertion process. In contrast to this investigation, Markovic et al. [21] captured the temperature at the surface of a bone segment. The two bone windows in this study allowed temperature to be measured directly on the implant surface. As a result, system-related measurement distortions due to differently positioned temperature probes in peri-implant bone could be avoided [9,10,13,14,15,16].

The above factors resulted in a relatively small variation of the temperature values per experimental group (Figure 6 and Figure 7). Thus, the goal of good reproducibility of the measurement conditions was achieved in the experimental setup used. In addition, the direct measurement of the surface temperature of the implants makes the nonsimulated, cooling effect of a well-perfused bone marrow smaller than with probes placed in the peri-implant bone [9,10,13,14,15,16]. Furthermore, the low thermal conductivity of bone (approximately 1/100 of titanium) can lead to thermal isolation of the sensors. The temperature-controlled thermo box (Figure 3(1)) additionally contributed to the good reproducibility of the measured temperature values (Figure 7).

In the selection of bovine ribs, care was taken to ensure that the layer thickness of the cortical bone did not differ a great deal (Figure 5b). It is apparent that, in a cylindrical implant, the heat generated by friction is primarily generated by the hard cortical bone and only slightly by the soft cancellous bone located below. The torques achieved during the insertion process were another indicator of identical experimental conditions: trial implants were hardly distinguishable in macro- and microdesigns and comparable layer thicknesses of the cortical bone (Figure 5a). In this study, the experimental parameters were chosen to simulate a high insertion torque (40–60 Ncm). It was above the average level suggested by most manufacturers (35 or 45 Ncm); nevertheless, this range was still below the maximum endurance for the ceramic implants.

In the case of rare outliers in insertion torques, the temperature readings were not included in the evaluation, but the complete measurement was repeated with the respective implant.

The remaining parameters (ambient temperature, rpm and feed at insertion, and dimension of the bone cavity) were kept reproducible and constant with conventional control circuits in the experimental setup (Figure 3).

### 4.3. Temperature Increase at Bone Window 1

The implant surface temperatures just below the cortical bone were highly significantly different between the ceramic and titanium implants. Statistical evaluation was performed using a group-adjusted linear mixed-effects model.

The increased temperature of the ceramic implant was due to its poor thermal conductivity. It was 2.5 W/mK and, thus, nearly ten times lower than titanium (22 W/mK). The thermal energy generated at the bone–implant interface during insertion dissipated poorly over the ceramic implant at locations further away. The thermal energy remained at the place of its formation and led to a higher temperature. This phenomenon is visible in Figure 7 from the infrared camera image. During the insertion process, the ceramic implant outside the bone (33.6 °C) heated up significantly less than the titanium implant (38.0 °C) under nearly identical experimental conditions. This means that much more heat (48.7 °C) was created in the bone at window 1 on the ceramic implant surface than in the titanium implant (42.9 °C). The increase in temperature, due to the greatly reduced thermal conductivity of ceramics, can hardly be reduced by intensive water cooling because the water has no access to the implant–bone interface during screwing-in of the implant [8], and the poor heat conduction of ceramics only cools the implant, which is not yet in the bone. It is obvious that only an extremely slow insertion of the implant can reduce the heating of the cortex bone [7,8,12,22].

Analysis of the temperature in bone window 1 (Figure 6) for all six experimental groups again showed a statistically significant increase in temperature at the ceramic implant compared to the titanium implant, regardless of the surface structure. However, the increase in temperature of ceramic implants with sandblasted and acid-etched surfaces (15.44 °C) was statistically significantly smaller than that with the machined (19.94 °C) or purely sandblasted (19.39 °C) ceramic surface. This effect of surface modification could possibly occur as a result of the high porosity of the sandblasted and acid-etched surface of ZrO_2_ implants (Figure 2). In particular, it shows nanoscopic pores smaller than 500 nm. This type of pore is too small to be reflected in the roughness average (Sa) but can be filled with blood and other tissue fluids, which in turn can function as heat storage and a cooling fluid. In other words, the blood and tissue fluid on the surface are heated. The heat capacity of water is 4.182 kJ/kg/K, which is more than 10 times higher than that of zirconia at 0.4 kJ/kg/K, and the density of zirconia is 6.08 g/cm^3^, which is 6 times higher than that of water. Thus, in order to increase the temperature of blood or tissue fluid on the blasted and acid-etched surface by 1 °C, a 1.72 times higher heat energy would be required than in the ZrO_2_ surfaces without fluids captured in nanoscopic pores—like machined (group 1) or sandblasted surfaces (group 2), where the latter results in a higher temperature increase at the implant–bone interfaces (Figure 6).

Another explanation of the result is based on a tribological effect [23]. Improved lubrication by the fluid trapped in the nanoscopic pores produces reduced heat development as it rotates through the cortical bone.

However, the temperature increase in the titanium implants is independent of their surface condition (Figure 6). This result might be related to good thermal conductivity, which allows rapid cooling of the titanium surface in bone window 1. In the bone window itself, there is no friction caused by close-fitting peri-implant bone.

### 4.4. Temperature Increase at Bone Window 2

The temperatures at the implant surface occurring in cancellous bone were different between the ceramic and titanium implants. Statistical evaluation was performed using a group-adjusted linear mixed-effects model.

The inter-group comparison (Figure 8) shows no statistically significant differences in temperature increase with respect to the test parameters of implant material and surface texture. Owing to low rigidity and the tissue structure interspersed with fat, the cancellous bone causes considerably less friction at the implant–bone interface than at the stiff and hard cortical bone. Therefore, the different thermal conductivities of the ceramic and titanium implants probably play a minor role in the temperature increase in bone window 2.

### 4.5. Differences in Bone Window 1 versus Bone Window 2

In this study, the bone windows for heat measurement were prepared 3 and 9 mm subcrestally in order to measure the change of the bone–implant interfacial temperature during implant insertion at the positions near the cortex and inside the implant cavity, respectively. As expected, the interfacial increase of temperature near the cortex was higher than that deep in the cancellous bone, regardless of the surface modification. In the study by Sener et al., the highest temperature in the process of preparing the implant bed was also observed in the superficial part of the implant cavity, and the heat decreased in the direction of the implant apex [17].

The different structure of cortical and cancellous bone as well as the higher frictional coefficient of hard cortical bone can lead to different frictional effects on the bone–implant surface; furthermore, the bone region at window 1 is exposed to friction for a longer time than the region of window 2 near the apex of the implant cavity. These factors, combined with the different thermal conductivities of the cortical and cancellous bone, result in more heat energy produced and accumulated in the cortical part during implant insertion.

Previously published studies showed different temperature increases during insertion of a titanium implant (0.55–9.81 °C) [12,21,24,25]. This might be due to temperatures being measured at the outer surface of the bone segment [24] or at a distance of 0.5–1 mm from the implant [12,21,25]. In addition, Sumer et al. [9] found more heat was generated with ceramic drills than with stainless steel drills at the superficial part of the drilling cavity.

### 4.6. Heat Caused Damage to Peri-implant Bone

The damage to the bone caused by overheating is related to the time it is exposed to the heating [26,27,28]. The higher the bone–implant interfacial temperature, the shorter the time it takes for bone damage to appear [26]. Eriksson and Albrektsson claim that bone damage occurs when the bone–implant interfacial temperature reaches at least 47 °C for 1 min [27]. As a result, the primary stability of the implants would decrease, and implants might loosen shortly after loading [28]. Furthermore, it was shown by Lundskog [26] that osteocytes underwent necrosis as soon as the bone was exposed to 50 °C for more than 30 s. Schmelzeisen et al. [28] showed that a temperature between 50 and 60 °C caused irreversible damage to osteocytes for an exposure time of 8 to 20 s. Based on that study, the median values of 55 °C and 14 s were applied as critical parameters in the current study. The interfacial temperature of ceramic implants measured near the cortical plate (bone window 1) were 56.94 °C on machined, 56.39 °C on sandblasted, and 52.43 °C on sandblasted-etched implants, with the starting temperature of 37 °C. Thus, the potential for bone damage induced by overheating was present on each surface. However, the heat and the relative exposure time decreased with the sandblasted and acid-etched surfaces, suggesting that the risk of bone damage could be reduced with proper surface modification.

Nevertheless, the results of this in vitro study might not fully represent the reality under clinical conditions [11,29]. The difference resulting from blood circulation and the thermal conductivity between nonlive and live tissues can influence the change in interfacial temperature [26]. Since the blood flow is six times higher in cancellous bone than in cortical bone, and blood can absorb part of the heat produced during implant insertion, the interfacial temperature increase in the cancellous bone is supposed to be smaller than that in the cortical bone. In view of this fact, the authors did not expect a significant cooling effect inside a patient’s cortical bone.

Based on this study, the heat produced during the implant insertion process mainly depends on the implant material and less on surface modification.

## 5. Conclusions

The results of this in vitro study show a temperature rise that is dangerous for the peri-implant cortical bone when a ceramic implant is inserted. Despite limited transferability to the clinical situation, the authors recommend a very slow insertion velocity for ceramic implants to avoid early implant losses.

## Figures and Tables

**Figure 1 jcm-08-01541-f001:**
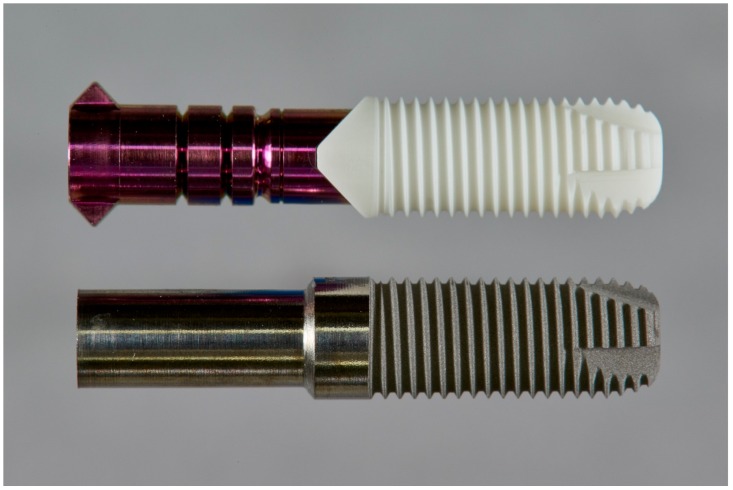
Implants made of zirconia and titanium with identical macrodesigns of the endosteal part coupled with the metallic insertion device.

**Figure 2 jcm-08-01541-f002:**
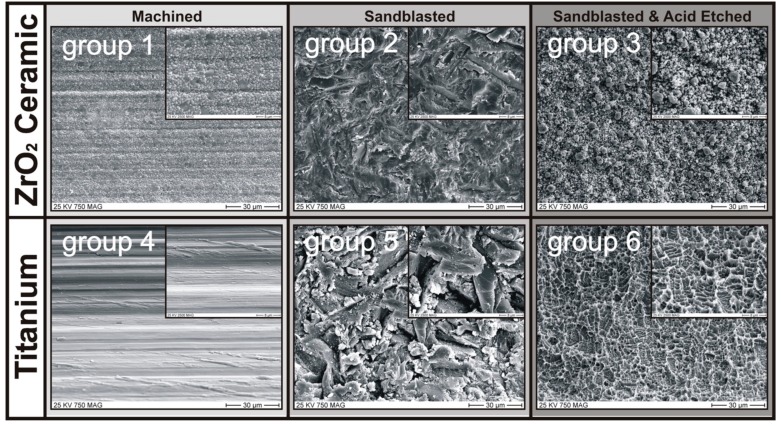
SEM images showing results of the different implant surface treatments. **Groups 1** and **4**: nontreated machined surfaces; **Groups 2** and **5**: sandblasted surfaces (corundum mesh size = 110 µm, working pressure = 6 bar); **Groups 3** and **6**: sandblasted and acid-etched surfaces (result: hydrophilic).

**Figure 3 jcm-08-01541-f003:**
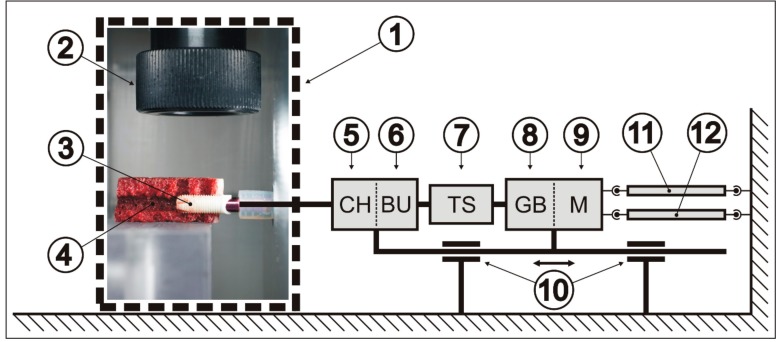
Diagram of the experimental setting. (**1**) Thermo box; (**2**) lens of infrared camera; (**3**) fully inserted ceramic implant; (**4**) transection of the rib segment; (**5**) collet holder (CH); (**6**) bearing unit (BU); (**7**) torque sensor (TS); (**8**) gearbox (GB); (**9**) motor (M); (**10**) linear bearing; (**11**) pneumatic cylinder; (**12**) distance sensor.

**Figure 4 jcm-08-01541-f004:**
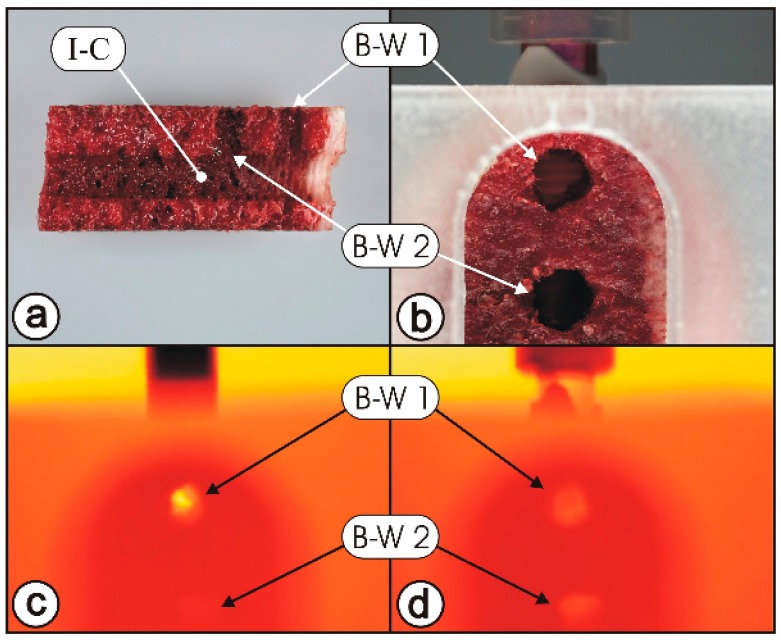
Dynamic temperature record at the bone–implant interface. (**a**) Transection of a rib segment; (**b**) overview of a rib segment; (**c**) infrared image at T3 in Figure 5; (**d**) infrared image at T7 in Figure 5. I-C = Implant Cavity 1; B-W 1 = Bone Window 1; B-W 2 = Bone Window 2.

**Figure 5 jcm-08-01541-f005:**
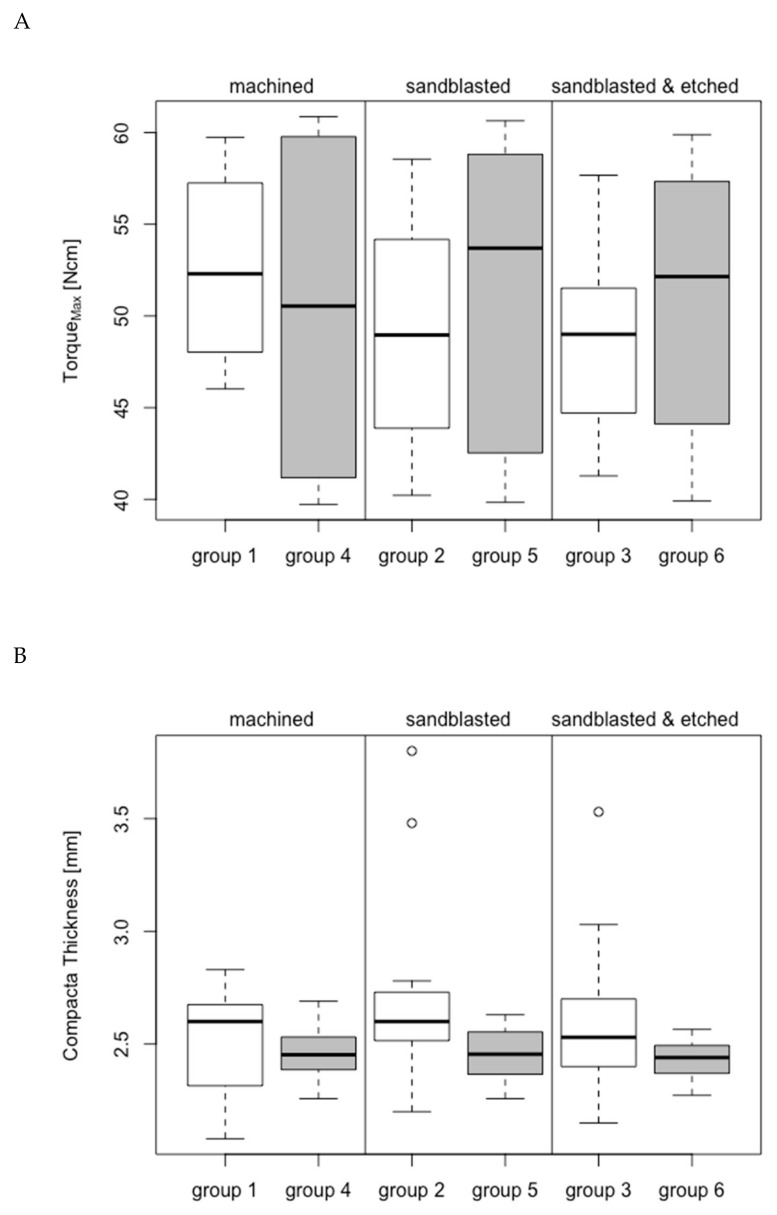
Box plots comparing (**A**) insertion torque, (**B**) cortical bone thickness, and (**C**) roughness.

**Figure 6 jcm-08-01541-f006:**
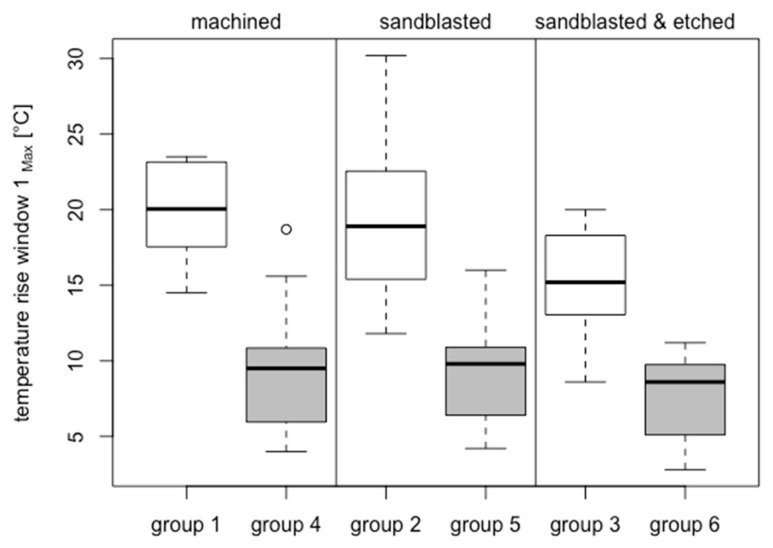
Box plots showing rise of temperature at bone window 1—results of all 6 groups.

**Figure 7 jcm-08-01541-f007:**
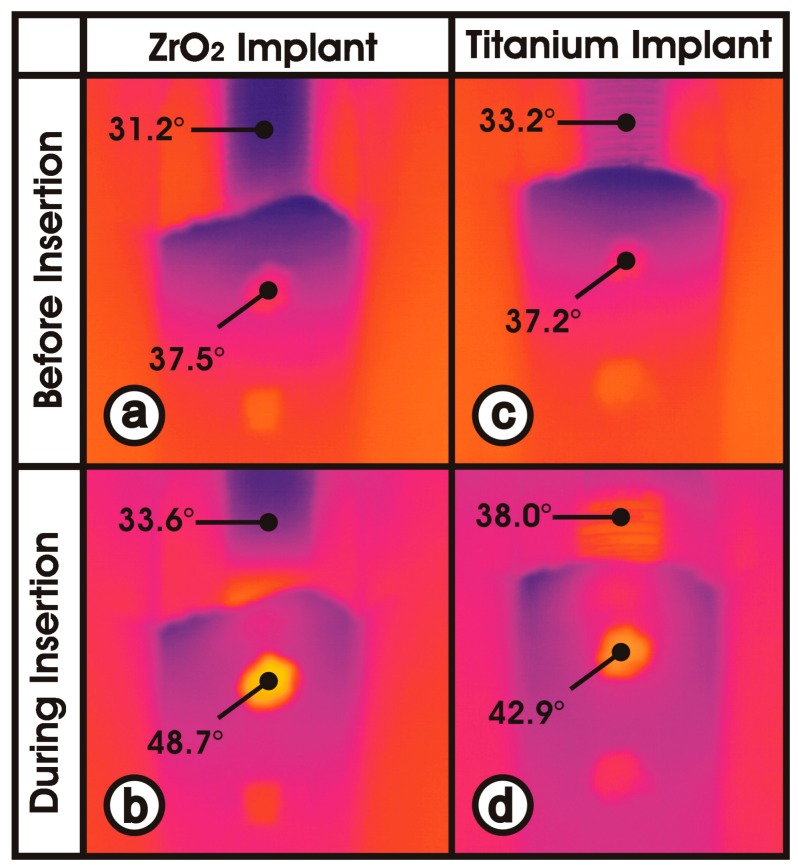
Material-dependent heat dissipation. (**a**) ceramic implant before insertion; (**b**) ceramic implant during insertion; (**c**) titanium implant before insertion; (**d**) titanium implant during insertion.

**Figure 8 jcm-08-01541-f008:**
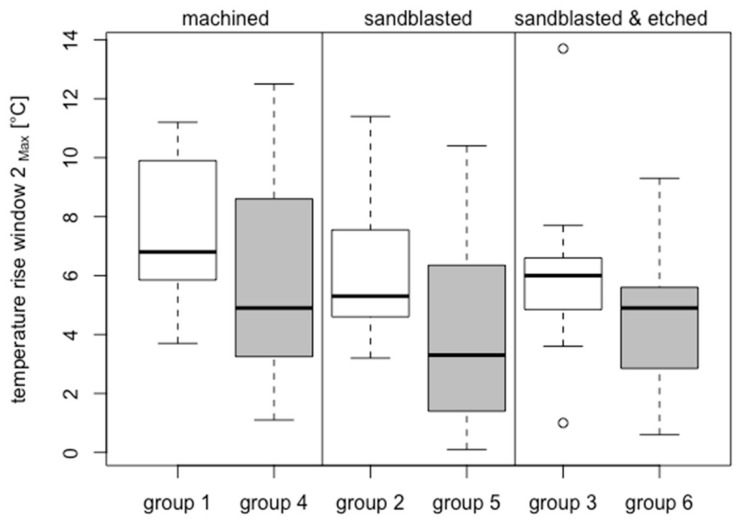
Box plots showing rise of temperature at bone window 2—results of all 6 groups.

**Table 1 jcm-08-01541-t001:** Temperature increase of the 6 groups.

Group	Analyzed (40–60 Ncm)/Inserted	Material	Surface Treatment	Surface Roughness Sa (µm)	Compacta Thickness (mm)	Insertion Torque (Ncm)	Temperature Increase BW-1 (°C)	ΔT > 10 °C BW-1		ΔT > 13 °C BW-1		ΔT > 18 °C BW-1		Temperature Increase BW-2 (°C)
								No.	Time (s)	No.	Time (s)	No.	Time (s)	
1	8/14	ZrO_2_	M	0.93 ± 0.16	2.51 ± 0.25	52.61 ± 5.10	19.94 ± 3.28	8/8	39.06	8/8	25.10	5/8	11.36	7.50 ± 2.64 °C
2	11/14	ZrO_2_	S	1.86 ± 0.38	2.75 ± 0.47	48.97 ± 6.43	19.39 ± 5.73	11/11	28.10	10/11	16.93	6/11	7.98	6.14 ± 2.40 °C
3	11/14	ZrO_2_	SAE	2.14 ± 0.62	2.61 ± 0.39	48.79 ± 5.19	15.44 ± 3.63	10/11	20.37	8/11	10.19	3/11	0.43	6.07 ± 3.11 °C
4	15/22	Ti Grade 4	M	0.37 ± 0.04	2.46 ± 0.12	50.10 ± 8.69	9.33 ± 4.18	7/15	22.49	3/15	19.07	1/15	20.40	6.03 ± 3.69 °C
5	15/20	Ti Grade 4	S	2.15 ± 0.24	2.46 ± 0.12	50.99 ± 7.97	9.20 ± 3.31	6/15	23.05	2/15	15.65	-	-	4.33 ± 3.26 °C
6	15/21	Ti Grade 4	SAE	1.89 ± 0.18	2.42 ± 0.09	50.72 ± 7.22	7.68 ± 2.68	2/15	17.45	-	-	-	-	4.41 ± 2.47 °C

M = machined, S = sandblasted, SAE = sandblasted and acid etched, BW = bone window.

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
