# Peer review of "Heat Generation at the Implant–Bone Interface by Insertion of Ceramic and Titanium Implants"

_jcm, 2019, doi:10.3390/jcm8101541_

Round 1

Reviewer 1 Report

Overall, the study was carefully conducted and the results seem valid and are presented sufficiently. I have few major and several minor remarks:

lines 51-52: change 'patients always opt for or clinicians always tend to use' to 'patients always opt for- or clinicians always tend to use'

lines 55: change 'This phenomenon described as aseptic loosening is caused' to 'This phenomenon, described as aseptic loosening, is caused'

lines 83, 84, etc: add city between the name of the manufacturer and its country

line 107: write out what FPA stands for

lines 114-115: a citation is lacking to support the statement that the bovine rib has similar properties/behaviour to human jaw

lines 113-118 (major remark): please elaborate why no bone mineral density measurement was performed; what would have been the outcome of the study if there was strong variation in specimen density?

lines 164-165: I don't understand the importance of this statement; did you use more then 14 Ti implants?

lines 165-166 and 195-196: you do explain this later on in the discussion section, however at this point, it is not clear why you only used data from 40 Ncm to 60 Ncm

lines 241-244: you found a significant difference in temperature increase for groups 2&3 and 1&3, so it does not seem adequate to pool these data and what was the purpose of this. Advice with the statistician.

lines 467-469: while I believe, that it is a valid point, that a lower velocity will result in less heat, this was not a direct finding of this study. I recommend to either remove this statement OR to discuss this first with some references in the discussion section. How would the change of the macro-geometry (e.g. larger thread) influence the results, what dothe authors think about it?

GENERAL COMMENTS:

I believe the data (results) presentation would have been much clearer in a table.

Indicate how many specimens in each group had to be excluded.

(major remark): The accuracy of the measurement of +/-2°C suggests an error of approx. 10% for the given results. While I am aware of the difficulty how to achieve a higher accuracy, the accuracy should be taken into account in statistical analysis.

One third of the references was published in the early 70s or early 80s. Was there no more recent literature?

Author Response

Dear reviewer,

please find our comments below. The revisions are inside the new manuscript version.

lines 51-52: change 'patients always opt for or clinicians always tend to use' to 'patients always opt for- or clinicians always tend to use'

Done.

lines 55: change 'This phenomenon described as aseptic loosening is caused' to 'This phenomenon, described as aseptic loosening, is caused'

Done.

lines 83, 84, etc: add city between the name of the manufacturer and its country

Done.

line 107: write out what FPA stands for

Done.

lines 114-115: a citation is lacking to support the statement that the bovine rib has similar properties/behaviour to human jaw

We included some literature [25-28].

lines 113-118 (major remark): please elaborate why no bone mineral density measurement was performed; what would have been the outcome of the study if there was strong variation in specimen density?

We explained in the manuscript.

lines 164-165: I don't understand the importance of this statement; did you use more then 14 Ti implants?

We explained in the manuscript and added table 1.

lines 165-166 and 195-196: you do explain this later on in the discussion section, however at this point, it is not clear why you only used data from 40 Ncm to 60 Ncm

Done.

lines 241-244: you found a significant difference in temperature increase for groups 2&3 and 1&3, so it does not seem adequate to pool these data and what was the purpose of this. Advice with the statistician.

Thank you for this comment. Indeed, it is not adequate compared. For comparison of zirconia vs titanium, we are now using a group adjusted linear mixed effects model to account for different methods.

lines 467-469: while I believe, that it is a valid point, that a lower velocity will result in less heat, this was not a direct finding of this study. I recommend to either remove this statement OR to discuss this first with some references in the discussion section. How would the change of the macro-geometry (e.g. larger thread) influence the results, what dothe authors think about it?

Thank you very much for this valuable hint. We have now discussed this point in detail in the section "Discussion" (see lines 476-480) and two articles (cited as no. 29 and no. 30) are added for the scientifically based justification of our thesis that only a reduction of the screwing speed can reduce the temperature rise at cortical bone.

GENERAL COMMENTS:

I believe the data (results) presentation would have been much clearer in a table.

We included table 1.

Indicate how many specimens in each group had to be excluded.

Is shown in table 1.

(major remark): The accuracy of the measurement of +/-2°C suggests an error of approx. 10% for the given results. While I am aware of the difficulty how to achieve a higher accuracy, the accuracy should be taken into account in statistical analysis.

The thermal camera system of the PI 160 has absolute and maximal accuracy of ± 2°C. Means we are not able to say if the absolute temperature is 37 °C (35 °C to 39 °C). But the sensitivity respectively the relative accuracy is 0.1 °C (0.1 K). This means if the temperature increase is 10 °C we are not able to say if it is from 37 °C to 47 °C. It also could be 35 °C to 45 °C or 39 °C to 49 °C. But the delta of 10 C° temperature increase is with an accuracy of 0.1 °C. The thermal energy which is needed to increase the temperature from 35°C to 45°C or from 39°C to 49°C is the same. I didn´t explained correctly in manuscript. I added the words “respectively the relative accuracy” in line 119 – 120.

One third of the references was published in the early 70s or early 80s. Was there no more recent literature?

Thank you for this comment. We add more recent literature, see cited 7 articles no. 24-30.

Reviewer 2 Report

This is a well conducted in vitro study, however some issues are present.

Why changing surfaces and not macro-morphologies (conical vs. cylindrical) or bone osteotomy diameter?  This should have made the study more “clinically interesting”

For example, this is a recent paper published on JCM: Intraosseous Temperature Change during Installation of Dental Implants with Two Different Surfaces and Different Drilling Protocols: An In Vivo Study in Sheep.

Line 196: why excluding torque under 40 N? Clinically even 30N could be considered a right value… even for immediate loading

Enossal? I would suggest to change it with endosteal

Author Response

Dear reviewer,

please find our comments below. The revisions are inside the new manuscript version.

Best regards and thank you for reviewing our article.

Why changing surfaces and not macro-morphologies (conical vs. cylindrical) or bone osteotomy diameter?  This should have made the study more “clinically interesting”

For example, this is a recent paper published on JCM: Intraosseous Temperature Change during Installation of Dental Implants with Two Different Surfaces and Different Drilling Protocols: An In Vivo Study in Sheep.

The increasing demand for ceramic implants, the clinically observed effect of anti-septic loosening and the very different thermal conductivities of the materials titanium and ZrO2 were the reason for the authors view of the assumed higher temperature increase during insertion of ceramic implants compared to Ti-implants. In order to get a clear statement about this, it was necessary to keep the outer implant shape as well as the implant-site-preparation identical. In addition, the authors wanted to find out whether the surface modifications, which vary widely in the market, have an additional impact on thermal development. Of course, different macro-designs of implants have a great clinical influence on the temperature generation in the bone, in particular the design of the thread geometry and / or the implant body (cylindrical vs. conical) as well as the underpreparing of the implant-site. However, if these parameters lead to increased temperature in the bone, it can be assumed based on the results of our study that ceramic implants on the cortical bone will have a higher temperature rise due to the much lower thermal conductivity of ceramics compared to titanium. However, this assumption (hypothesis) has to be certainly proven in similar in vitro set-ups in future. The latter is also the plan of this group of authors.

Thank you for the new article. After reading carefully, we cited it.

Line 196: why excluding torque under 40 N? Clinically even 30N could be considered a right value… even for immediate loading

Thank you for your comment: We explained now in line 132 – 137 and in line 187-195.

Enossal? I would suggest to change it with endosteal

Done.
